# Haloperidol, Olanzapine, and Risperidone Induce Morphological Changes in an In Vitro Model of Human Hippocampal Neurogenesis

**DOI:** 10.3390/biom14060688

**Published:** 2024-06-13

**Authors:** Bálint Jezsó, Sára Kálmán, Kiara Gitta Farkas, Edit Hathy, Katalin Vincze, Dzsenifer Kovács-Schoblocher, Julianna Lilienberg, Csongor Tordai, Zsófia Nemoda, László Homolya, Ágota Apáti, János M. Réthelyi

**Affiliations:** 1Institute of Molecular Life Sciences, HUN-REN RCNS, Magyar tudósok körútja 2., H-1117 Budapest, Hungary; jezso.balint@ttk.elte.hu (B.J.);; 2Doctoral School of Biology, Institute of Biology, Eötvös Loránd University, Pázmány Péter sétány 1/c., H-1117 Budapest, Hungary; 3ELTE-MTA “Momentum” Motor Enzymology Research Group, Department of Biochemistry, Eötvös Loránd University, Pázmány Péter sétány 1/c., H-1117 Budapest, Hungary; 4Albert Szent-Györgyi Health Centre, Department of Psychiatry, University of Szeged, Szentháromság utca 5., H-6722 Szeged, Hungary; kalmansara@gmail.com; 5Department of Psychiatry and Psychotherapy, Semmelweis University, Balassa utca 6., H-1083 Budapest, Hungary

**Keywords:** human induced pluripotent stem cells, neural differentiation, hippocampal neurogenesis, antipsychotics, neurite outgrowth

## Abstract

Background: Induced pluripotent stem cell (iPSC) based neuronal differentiation is valuable for studying neuropsychiatric disorders and pharmacological mechanisms at the cellular level. We aimed to examine the effects of typical and atypical antipsychotics on human iPSC-derived neural progenitor cells (NPCs). Methods: Proliferation and neurite outgrowth were measured by live cell imaging, and gene expression levels related to neuronal identity were analyzed by RT-QPCR and immunocytochemistry during differentiation into hippocampal dentate gyrus granule cells following treatment of low- and high-dose antipsychotics (haloperidol, olanzapine, and risperidone). Results: Antipsychotics did not modify the growth properties of NPCs after 3 days of treatment. However, the characteristics of neurite outgrowth changed significantly in response to haloperidol and olanzapine. After three weeks of differentiation, mRNA expression levels of the selected neuronal markers increased (except for MAP2), while antipsychotics caused only subtle changes. Additionally, we found no changes in MAP2 or GFAP protein expression levels as a result of antipsychotic treatment. Conclusions: Altogether, antipsychotic medications promoted neurogenesis in vitro by influencing neurite outgrowth rather than changing cell survival or gene expression. This study provides insights into the effects of antipsychotics on neuronal differentiation and highlights the importance of considering neurite outgrowth as a potential target of action.

## 1. Introduction

Psychiatric disorders affect more than 1 billion people worldwide, account for approximately 7% of all global burden of disease, rank fifth globally in causes of disability [1], and necessitate psychopharmacological treatment in many cases. Our understanding of mental health conditions is fragmented, and their examination still faces many obstacles. Although cutting-edge neuroimaging techniques, such as high-resolution functional magnetic resonance imaging [2] or, in the case of animal models, two-photon microscopy, can provide a relatively high spatiotemporal resolution of underlying neural mechanisms [3], these techniques are not widely available. In addition, in vivo sampling from the human brain is mostly inaccessible, and post-mortem tissue samples are not appropriate for functional assays. Mental disorders are considered exclusively human conditions; therefore, investigating and modelling these conditions in animals raises several problems and necessitates compromises. The above limitations affect psychopharmacological research as well: little is known about the effects of psychoactive agents at the level of human neural and glial cells, neural circuits, neurogenesis, and neuronal differentiation. Cell cultures, especially patient-derived cell lines, are promising alternative model systems in translational psychiatry and drug development and can delineate a more detailed picture of the above phenomena.

Human induced pluripotent stem cells (iPSCs) and their progeny are considered important tools for in vitro disease modelling and drug development. Remarkably, a combination of growth factors and small molecules during the differentiation of iPSCs is sufficient to evoke not only neuronal cell fate but also high neurotransmitter specificity. mRNA expression profiling has shown that human iPSC-derived neural cells closely follow developmental programs from early embryogenesis to late fetal periods. In other words, neuronal progenitor cells (NPCs) differentiate in vitro into neuronal cells of various stages of maturity, recapitulating in vivo spatio-temporal neurodevelopmental trajectories [4,5,6].

Postnatal neurogenesis continues lifelong in the neurogenic niches of the human brain. The best-known area is the subgranular zone of the dentate gyrus (DG), from where the new, immature neurons arise and migrate towards the inner granule cell layer of DG, then differentiate into mature, prospero homeobox 1 (PROX1) positive glutamatergic granule cells. This dynamic, finely-tuned process is crucial for episodic and spatial memory and for the process called pattern separation, which is defined as the transformation of non-simultaneous but similar input patterns into more distinct output patterns [7,8,9,10]. In various psychiatric disorders, including schizophrenia and major depressive disorder, the neurogenic properties of the DG have been shown to decline (reviewed by [11,12]). It is not easy to estimate to what extent these changes can be attributed to the course of the disease or to the unwanted side effects of pharmacotherapy. Some longitudinal clinical and experimental studies reported that chronic treatment with haloperidol (HP), a typical or first-generation antipsychotic, and olanzapine (OL), an atypical or second-generation antipsychotic medication, is associated with significant shrinkage of grey matter [13,14,15,16]. In contrast, some antidepressants and antipsychotics (including HP and OL) have demonstrated neuroprotective features as well. Some of these medications might exert their therapeutic effects partially by improving neurogenesis (reviewed by [12,17]).

Several aspects of schizophrenia, which are related only indirect to neuronal function, could be a target of pharmacological treatment. For instance, oxidative stress has been implicated, as the increased enzyme activity of superoxide dismutase and plasma glutathione peroxidase, the elevated plasma concentration of malondialdehyde, and other markers of oxidative stress have been found to correlate with clinical phenotypes [18]. Moreover, the expression level of Brain-Derived Neurotrophic Factor (BDNF) decreased significantly in schizophrenia patients and showed a negative correlation with serum levels of oxidative stress markers [18,19]. As some studies reported, atypical antipsychotics can increase the patient’s serum concentration of BDNF [20,21,22]; some of them have an enhancer effect on the *BDNF* promoter [23] and may have an antioxidant effect as well [24]. In murine and primate model systems, in situ hybridization studies revealed an increased level of BDNF in the hippocampal CA1, CA3, and DG following OL or clozapine treatment [25], but opposite effects were observed in the case of HP in the hippocampus, hypothalamus, striatum, and prefrontal cortex [25,26,27,28]. Accordingly, some studies reported increased neurogenesis in the hippocampus after treatment with the atypical (second-generation) antipsychotics OL, clozapine, quetiapine, and risperidone (RP), based on BrdU labeling, but HP had no such effect [29,30]. In contrast, a significant decrease was observed in the number of doublecortin-positive cells in the murine hypothalamus following HP administration [31], suggesting that HP has no effect on the proliferation of NPCs but impairs their survival, in line with the diminished BDNF level mentioned previously. Taken together, first- and second-generation antipsychotics can influence the neurogenic activity in vivo, most probably via changes in BDNF expression or other factors of the brain’s stem cell niches.

Another interesting aspect of schizophrenia is related to the innate immune system. Serum concentrations of several inflammatory mediators (such as IL-1β, IL-6, IL-8, IL-12, TGF-β, TNF-α, INF-γ) are increased, some of them positively correlate with symptom severity, while others respond to medication, indicative of a direct or indirect interaction between inflammation and antipsychotic treatment. The main cellular representative of the immune system in the CNS is microglia, which are involved in many developmental and maturational processes of the brain via synaptic pruning. A widely accepted hypothesis formulates that inappropriate pruning contributes to the pathomechanism of schizophrenia, and it might relate to the elevated level of cytokines and inflammatory factors. Since dopamine receptors D1 and D2 are expressed by microglia as well, the dopamine-antagonist psychotropic agents can modulate microglial activity and may exert a favorable effect for patients by this mechanism as well [32,33,34,35].

Antipsychotics affect neural functions in a multidimensional manner; however, little is known about the direct effect of these psychotropic medications on proliferating neuronal progenitors and neural differentiation. In 2014, Yu et al. published a protocol for the differentiation of hippocampal DG glutamatergic granule neurons from human iPSCs based on key developmental cues specific to DG identity [36]. This paradigm opened an avenue for studying DG neurogenesis in vitro and was successfully used to reveal transcriptomic and functional alterations in neuronal cells derived from patients living with schizophrenia [36,37].

In the current study, we aimed to investigate how dopamine and serotonin antagonist antipsychotics affect the proliferation of progenitor cells, neurite development and gene expression of differentiating hippocampal granule cells in vitro. We treated the iPSC-derived neural progenitor cells with a dopamine receptor 2 (D2) antagonist haloperidol (HP), a D2 and serotonin 2A (5-HT2A) antagonist olanzapine (OL), and a D2, 5-HT2A, and alpha-2 adrenoceptor antagonist risperidone (RP). These antipsychotics were chosen based on their widespread and long usage and high market share in the pharmacological treatment of schizophrenia and other forms of psychosis. In our experiments, the chosen drugs were applied in a therapeutically relevant dose based on patient serum concentrations [38,39,40]. A balanced set of genes was chosen for quantification of mRNA expression: transcription factor essential for lineage specification NeuroD1, glial and neuronal markers such as the astrocyte marker glial fibrillary acidic protein (GFAP), and microtubule-associated protein 2 (MAP2), and synaptic machinery proteins vesicular glutamate transporter (vGLUT1), metabotropic glutamate receptors (mGLUR2 and mGLUR7) characteristic for DG granule cells.

## 2. Materials and Methods

### 2.1. Cell Cultures

Human iPSCs were differentiated into PROX1-positive hippocampal granule cells following the protocol published by Yu and colleagues [36]. Briefly, iPSCs derived from fibroblasts of a female healthy control subject (line 6/2/F) described earlier [41] were transferred onto ultra-low adherence dishes for free-floating embryoid body (EB) formation. EBs were treated with N2B27 differentiating media and plated on poly-L-ornithine and laminin-coated plates on day 20. After 7–10 days, adhered neural rosettes containing EBs were manually collected. The originated neural precursor cells (NPCs) were passaged in N2B27 media plus basic Fibroblast Growth Factor2 (bFGF2) and laminin. NPCs between p5 and 15 were used for further experiments. For proliferation assay, the NPCs were cultured in N2B27 media plus bFGF2, and laminin and the pharmacological agents were added to this medium. The neural differentiation was initiated using BDNF, WNT family member 3A (WNT3A), cAMP, and ascorbic acid-containing medium, and treatments with HP, OL, and RP were made in this medium. The pharmacological agents were solved in dimethyl sulfoxide (DMSO) and added to the differentiation medium (Table 1). The control cells were treated with the same differentiation medium containing an equal amount of DMSO or nothing. The culturing media was changed three times per week for 21 days. All cell cultures were grown under standard conditions (37 °C, 5% CO_2_), and the media were supplemented with combined antibiotic-antimycotic.

### 2.2. Proliferation Assay

To perform the proliferation assay, we seeded 20,000 cells/well onto a 96-well plate. After 24 h, the culture medium was changed for antipsychotic supplemented (HP, OL, and RP, respectively) at two different concentrations (Table 1) and changed daily. Since some of the antipsychotics were dissolved in DMSO, the medium was supplemented with additional DMSO in all treatments; therefore, the final concentration of DMSO was the same in all treatment groups. The proliferation rate was quantified by cell nuclei counts at 24, 48 and 72 h after the start of the treatment. For this purpose, cells were incubated with Vybrant™ DyeCycle™ Violet (DCV) chromogenic nuclear stain (25 nM) for 1 h. After the medium exchange, the fluorescent signal was detected by ImageXpress^®^ Micro XLS High-Content Analysis System (Molecular Devices, San Jose, CA, USA) using a DAPI filter frame (ex. 377/50 nm, em. 447/60 nm) and a 10× Nikon (Nikon Instruments Inc., Melville, NY, USA) objective (Plan Fluor, NA = 0.3). Nine non-overlapping pictures were taken from every well. The data obtained from 3 independent experiments were analyzed by ANOVA with a post hoc Tukey test.

### 2.3. Neurite Outgrowth Assay

To perform neurite outgrowth experiments, we seeded 2.5 × 10^4^ cells per well onto eight-well chambers coated with poly-L-ornithine-laminin. Neurite length measurements were performed on day 10 after the initiation of neuronal differentiation with the additional antipsychotic or DMSO treatments. We used Calcein-AM with a final concentration of 100 nM for staining the cytoplasm and processes and DCV (25 nM) for nuclear staining. After 1 h incubation time, the medium was exchanged. To detect fluorescent signals, we used the ImageXpress^®^ Micro XLS microscopy system with DAPI (ex. 377/50 nm, em. 447/60 nm) and FITC (ex. 482/35 nm, em. 536/40 nm) filter frames and a 10× Nikon objective (Plan Fluor, NA = 0.3) as described earlier [42]. For data analysis and evaluation, the Neurite Outgrowth module of MetaXpress software (64-bit version number is 5.3.0.5, Molecular Devices) was performed. Data from 3 independent experiments were evaluated. For statistical analysis, the Kruskal–Wallis test and post hoc Dunn’s test were carried out.

### 2.4. Gene Expression Analysis with Real-Time Quantitative PCR

For real-time quantitative PCR analysis NPCs were plated to poly-L-ornithine and laminin coated 24 well plates. After 21 days of differentiation and antipsychotic treatment, total cell RNA was isolated with Trizol (Invitrogen, Carlsbad, CA, USA) following the manufacturer’s protocol. The quality of the RNA samples was assessed with a Nanodrop 2000 Spectrophotometer (Thermo Scientific, Waltham, MA, USA). Complementary DNA (cDNA) synthesis was carried out using the Promega Reverse Transcription System according to the manufacturer’s guidance.

The transcription of metabotropic glutamate receptor 2 (mGluR2); metabotropic glutamate receptor 7 (mGluR7); vesicular glutamate transporter 1 (VGLUT1 or SLC17A7); microtubule-associated protein 2 (MAP2); neuronal differentiation 1 (NeuroD1 or BETA2); and glial fibrillary acidic protein (GFAP) genes were measured with real-time quantitative PCR using TaqMan probes. The endogenous control was the large ribosomal protein P0 (RPLP0). The reactions were performed according to the manufacturer’s instructions, in triplicate for every well. mRNA changes were measured in the StepOnePlus real-time PCR System (Applied Biosystems, Waltham, MA, USA) using the StepOne v2.1 program. During statistical analysis, the Kruskal–Wallis test and post hoc Dunn’s test were performed.

### 2.5. Immunocytochemistry

For immunofluorescence staining, 2.5 × 10^4^ cells per well were plated onto eight-well chambers. Fixation and staining of the cells were performed on day 21 after initiation of neuronal differentiation with antipsychotic treatment. Fixation was performed with 4% paraformaldehyde (15 min) and washed with DPBS. Then, the sample was blocked and permeabilized with DPBS, containing 5% goat serum, 2 mg/mL bovine serum albumin (Sigma-Aldrich, St. Louis, MO, USA), 1% fish gelatin, and 0.1% Triton-X (complete blocker). The primary antibodies (anti-GFAP mouse, 1:100 dilute; anti-MAP2: monoclonal/mouse, 1:500 dilute, Sigma/Merck, Darmstadt, Germany) were incubated for 1 h. After washing with DPBS, the following secondary antibody was incubated, also for 1 h: Alexa Fluor 568 goat anti-mouse IgG (Life Technologies, Carlsbad, CA, USA). After washing again with DPBS, nuclei were labelled with Hoechst 33342 DNA stain. Neurons were examined under a Zeiss (Oberkochen, Germany) LSM 710 confocal microscope (20× pan-apochromatic objective, NA = 0.8). Data were collected from 5 to 10 non-overlapping pictures from each cell culture of 4 independent experiments. The image analysis was performed using the ImageJ software package (downloaded in 2022), measuring the integrated pixel density of the MAP2 and GFAP immunostaining, respectively. The fluorescent value of each picture was normalized by the number of recorded cells determined through the thresholding method. To avoid sampling bias due to the different number of images from each experiment, for statistical analysis intensity values from the same experiment were averaged. Since the Shapiro–Wilk test indicated normal distribution in all treatment groups, ANOVA was performed with the post hoc Tukey test.

## 3. Results

Our primary question was how antipsychotics affect the physiology of hippocampal progenitor cells. Therefore, we first examined whether the used antipsychotics affect the growth of NPCs. This test was performed during 3 days of treatment. Next, we examined the initial stage of neuronal differentiation; we started differentiation in the direction of DG neurons and assessed the effect of antipsychotics on neurite outgrowth at 10 days. Thereafter, in independent experiments, we continued the differentiation until 3 weeks to examine the expression of important genes at both the RNA and protein levels in order to monitor the effect of antipsychotics in differentiating neurons.

### 3.1. Characterization of Neural Cell Types

Based on the differentiation protocol described previously [36] and used several times in our laboratory [37,41,42,43], we first created neural progenitors from an iPSC line of a healthy donor individual. Among the neural progenitor markers, SRY-box transcription factor 2 (SOX2) and Nestin were used to characterize NPCs. Immunostainings showed that the cultures uniformly expressed both proteins (Figure 1a). After 5 weeks of further neuronal differentiation of NPCs, MAP2 and PROX1 immunostaining was used to characterize the neuronal culture. MAP2 is a general neuronal marker [44,45], while PROX1 is specific and essential for the survival and commitment of the DG neuronal progenitors and their progeny [46,47]. At the end of the differentiation, approximately 60% of the cells exhibited PROX1 and MAP2 positivity (Figure 1b), demonstrating the efficiency of the protocol.

### 3.2. Proliferation of Neural Progenitors

One of the characteristics of neuronal progenitors is their ability to multiply within a permissive environment. Deviations in the proliferation of NPCs may play an etiological role in several psychiatric disorders [48,49]. Therefore, we examined whether the growth rate of NPCs changes as a result of the antipsychotics applied at a low and a higher concentration (Figure 2). DMSO at the concentration of 0.2 µL/mL was also used as an additional control since the antipsychotics were dissolved in this reagent. After 72 h of culturing, we found that the cell number doubled in all cultures. Neither HP, OL, nor RP treatment induced significant alterations in the cell number of hippocampal NPC cultures.

### 3.3. Neurite Outgrowth Assay

At the beginning of neural differentiation, the progenitor cells start to grow non-specialized processes called neurites, which later form dendrites and axons in the mature polarized neurons. The process of neurite outgrowth can have a significant influence on the maturation of neurons, and its alterations can play a role in the development of disorders [50,51]. In this experiment, the neural progenitor cells were differentiated for 10 days with or without the addition of selected medications. We found that a low concentration of HP (10 ng/mL) significantly enhanced the neurite formation based on an elevated number of processes, neurite outgrowth and arborization, while it had an inhibitory effect on cell number compared to the control (Figure 3 and Appendix A). A low concentration of OL also showed beneficial effects on neurite outgrowth and arborization, showing the possible common effect of antipsychotics on neuronal development.

### 3.4. Gene Expression Alterations

We first examined how the mRNA expression of the selected genes changes during neural differentiation in untreated cells, comparing mRNA expression in NPCs and three-week-old neurons. We found that GFAP, mGluR2 NeuD1 and VGLUT1 increased significantly, mGluR2 showed only a slight, non-significant increase, while MAP2 mRNA level decreased slightly (Appendix A). Next, we investigated the effects of 21-day-long HP, OL and RP on the gene expression levels of differentiating dentate gyrus granule cells.

The relative mRNA levels of the studied genes compared to the DMSO-treated and non-treated control cells are presented in Figure 4. According to our data, only high concentration RP was able to significantly increase the mRNA expression of MAP2 (compared to DMSO treated controls) and mGluR2 and mGluR7 (compared to high concentration of HP). Basically, the antipsychotic treatment caused no major changes in the mRNA expression of selected neural genes.

### 3.5. Immunocytochemistry

Expression levels of mRNA are not always consistent with protein levels in the cell. Next, we wondered whether the relatively small variability of mRNA levels is also realized at the protein level. Therefore, the most highly expressed MAP2 (Figure 5a and Appendix A) and astrocyte-specific GFAP proteins (Figure 5b and Appendix A) were further examined using immunocytochemistry. We examined 5 to 10 image fields in 4 independent experiments and found that neither MAP2 nor GFAP protein levels were significantly different when comparing antipsychotic-treated and control samples. At such an early stage of differentiation, the cultures are still very heterogeneous; the maturation state and differentiation capacity of the cells are different, so the variance in the protein level is large. This is especially true for the few glial cells that are arranged in groups; therefore, the standard deviation is even larger between view fields.

## 4. Discussion

Over recent years, iPSC technology has emerged as a promising research tool in molecular psychiatry and drug testing [52,53]. For antipsychotics, various models are available to study pharmacological effects (reviewed by [54]); however, testing these drugs in human iPSC-derived cell lines remains a significant gap in the current literature. There is a scarcity of well-designed pharmacological studies testing the effects of antipsychotics in human iPSC-derived neurons [55]. The primary goal of our research was to demonstrate that in vitro iPSC-derived hippocampal granule cells are valuable tools for studying psychotropic drug mechanisms. Therefore, we investigated the direct effects of first- and second-generation antipsychotics on the proliferation of iPSC-derived neural progenitors, as well as neuronal development of differentiating hippocampal granule cells in vitro. Our results indicated an effect of haloperidol and olanzapine exposition on neurite outgrowth kinetics, but no changes in the expression of key neuronal genes nor in short-term cell survival were observed.

Antipsychotic medications exert their effects on specific receptors; for example, HP is a D2 antagonist, while OL targets both D2 and 5-HT2A receptors, and RP acts on D2, 5-HT2A, and alpha-2 adrenergic receptors [56]. Asada and colleagues reported that most of the iPSC-derived neural stem cells do not express D2 receptors, yet they observed a change in differentiation capacity in response to these medications. The authors hypothesized that antipsychotics exert their effects on alternative or noncanonical pathways [57]. Based on unpublished data from our laboratory from mRNA sequencing, the hippocampal neural progenitors and differentiating neuronal cells express a variety of receptors that are targeted by antipsychotics, including D2, D4, 5-HT2A, 5-HT2C, 5-HT6, and 5-HT7 receptors (Appendix A). Based on this information, we can hypothesize that these drugs operate both at pharmacological target receptors and through other noncanonical ways in iPSC-derived neuronal cultures.

In mouse embryonic hippocampal cultures, Osacka et al. showed that 72-h treatment with HP and OL inhibit proliferation both at 0.1 μM and 10 μM concentrations [58], which may also be attributed to the cytotoxic effect of these drugs and their metabolites [59,60]. In contrast, Heiser et al. found that OL can support the growth of the SH-SY5Y human neural cell line and only inhibits proliferation [61]. In comparison, in the investigated human iPSC-derived NPC cultures, we did not observe any changes in growth curves under HP or OL treatment. Regarding risperidone, a mild cytotoxicity is also known from the literature, but in vitro this effect emerges at approximately 100 times higher concentrations compared to what we applied [62,63]. In agreement with this data, our results do not reflect any inhibitory effect of RP on proliferation (Figure 2). In our experimental setting, we applied concentrations of the antipsychotics adjusted to human serum levels, which are an order of magnitude lower than what has been shown to have short-term cytotoxic effects [61]. Although up to 10–30 times higher concentrations can be measured in the brain tissues of rats [64,65], there is no similar evidence in humans. Taking this into account, the antipsychotics were applied at a lower and a higher concentration, but we found no significant changes in the cell counts as a result of the higher concentration (closer to the previous studies).

It needs to be underscored that in the case of the 10-day treatment, the number of cells measured in the neurite outgrowth experiments was slightly reduced by HP. This result is consistent with previous data showing that long-term HP treatment has a neurotoxic effect [66,67], while this has not been confirmed for atypical antipsychotics [68]. In the neuronal differentiation experiments, developing neurons were exposed to a different set of external factors and signals compared to the initial NPC experiments. In this system, a 10-day-long treatment resulted in decreased cell count at both HP concentrations, but it is uncertain whether the active substance inhibits proliferation or perhaps induces apoptosis by increasing caspase activity. The induction of caspase activity due to HP treatment has previously been described in rat cortical [69] and human neuroblastoma cultures [70].

Neurites, processes originating from the cell body of differentiating neurons, play a crucial role in establishing connections between neurons. Deficits of neurite outgrowth can disrupt neural connectivity and impact brain function. It should be emphasized that neurites are not static structures and have the ability to change, forming the foundation for neuroplasticity throughout an individual’s lifespan [71]. There is compelling evidence indicating neurite deficits in models of severe mental disorders such as schizophrenia, autism spectrum disorder, Alzheimer’s disease, and major depression [72,73,74,75]. Thus, improving neurite outgrowth and enhancing neural connectivity may contribute to the therapeutic effects of antipsychotic medications. On the other hand, excessive administration of antipsychotics could potentially impair neurites and lead to adverse conditions. In vitro studies demonstrated controversial reports about axonal development in the presence of antipsychotics [76]. For example, no differences were found in neurite length in human neuroblastoma SH-SY5Y cells treated with HP for 4 days [77], but a later study showed shorter neurites after 3 days of both HP and OL treatment in mouse hippocampal neuronal cultures [58]. Interestingly, we observed opposite effects in our human cell culture, as HP promoted neurite development, while OL also positively influenced certain features. Apart from obvious species differences, variations in treatment duration and cell differentiation states could also account for these discrepancies. We observed an anticorrelation between neurite length and cell number in differentiating neural cultures (Spearman’s rank correlation obtained −0.568, regardless of the treatment regimen), indicating that changing the cell density could also be an important parameter and a potential source of the above-mentioned discrepancies.

The effect of antipsychotics on neurogenesis in vivo remains unclear [30,31,58]. The differences between species, the complex niche existing *in vivo*, the differences in the different brain areas examined in adult animals, and the applied doses and duration of treatment make simple comparisons of human pluripotent stem cell-based models challenging. Therefore, we will not discuss this part of the literature in detail here.

The expression of key neuronal genes was not significantly affected by antipsychotic treatment neither at mRNA, nor protein level. Of note, we investigated the beginning of hippocampal differentiation, using an *in vitro* model of adult human neurogenesis, therefore, synaptic function is immature at this stage. Animal models and *post mortem* studies have indicated that chronic treatment with atypical antipsychotics causes downregulation of mGluR2 via 5-HT2A receptor-mediated epigenetic regulation [78]. Changes in synaptic structure and/or function caused by drug treatment may occur at later time points, but this was beyond the scope of our study. To our surprise, we found that the MAP2 mRNA expression was higher in NPCs than in differentiating neurons (Appendix A). However, when we investigated MAP2 protein using ICC, this difference was not detectable at the protein level (Appendix A).

Limitations of stem cell-based models should be acknowledged. The two-dimensional structure is not ideal for characterizing cell–cell and cell–niche interactions, and it can only mimic the fetal state of neuronal development, while these psychotropic medications are administered to patients in adulthood. We used a cell line derived from a healthy donor, and throughout our experiments, we could not model the environmental stressors that NPCs are exposed to. Broadening our methodology by including samples from psychiatric patients and introducing three-dimensional culture conditions could provide additional valuable data about the applicability of stem cell-based disease models and the elucidation of drug mechanisms.

In summary, our data demonstrate that the effects of antipsychotics can be investigated in this human *in vitro* model system. We found that the short treatments mainly affected the development of neurites but not cell numbers or the differentiation ability of neuronal progenitors.

## Figures and Tables

**Figure 1 biomolecules-14-00688-f001:**
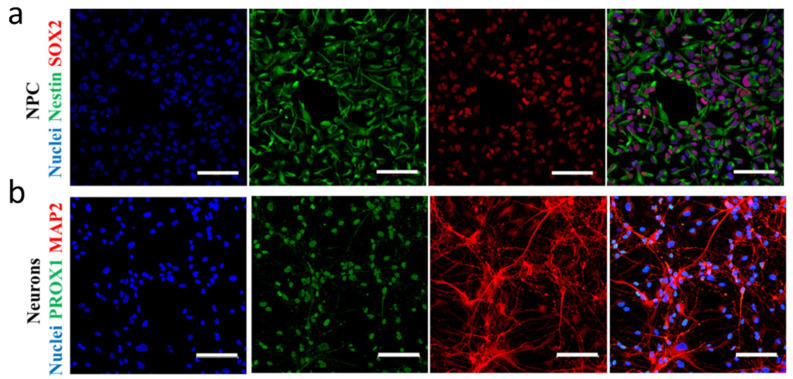
Immunostaining for characterization of neural progenitors and neurons. (**a**) Representative confocal images of Nestin (green) and SOX2 (red) progenitor markers NPCs. (**b**) Representative confocal images of PROX1 (green) and MAP2 (red) neural markers in differentiated neurons. The nuclei were counterstained by Hoechst 33342. The scale bars represent 100 µm.

**Figure 2 biomolecules-14-00688-f002:**
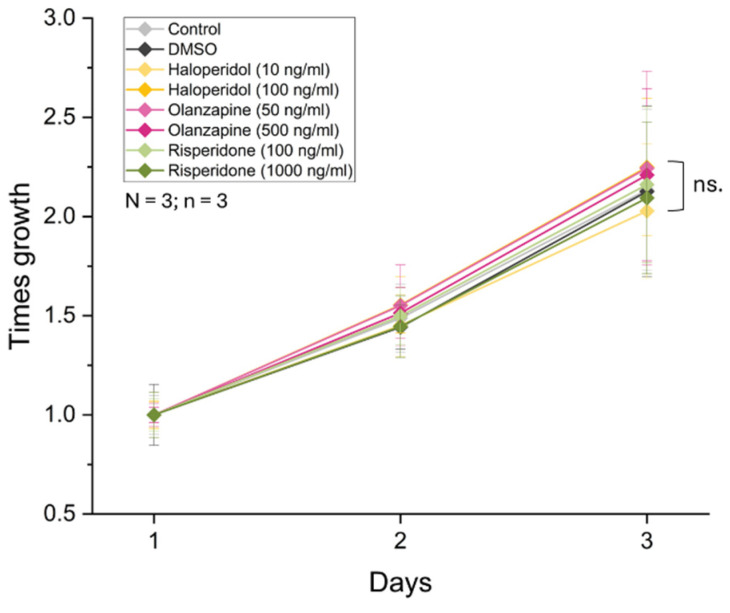
Growth curves of NPCs. The cells were stained with DCV and measured by high-content screening for 3 days. Data were normalized to first-day cell numbers. The graphs show the average of cell number ± SD at least three biological and 3-3 technical parallels. Data were collected from 9 non-overlapping view fields for each parallel. For statistical evaluation ANOVA was performed, with a post hoc Tukey test.

**Figure 3 biomolecules-14-00688-f003:**
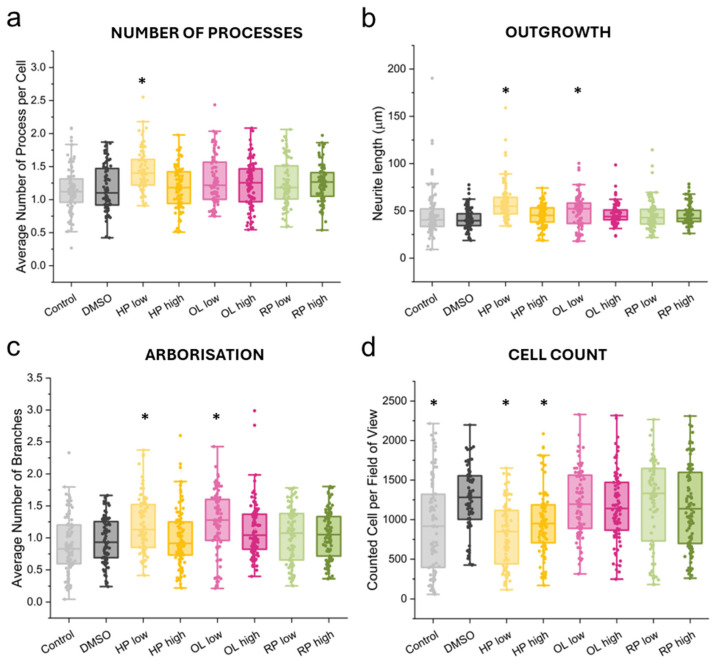
Neurite outgrowth analysis on the 10th day of neural differentiation. Analysis of neurite outgrowth characteristics: (**a**) average number, (**b**) length, and (**c**) arborization of neurite outgrowths. In panel (**d**), the number of examined cells was given, with which the characteristic data of a number of processes and neurite outgrowth were normalized. Kruskal–Wallis test and post hoc Dunn’s test have been performed (asterisk signs the significant difference to the DMSO treated control). Data were collected from three biological parallels and at least nine non-overlapping view fields for each parallel.

**Figure 4 biomolecules-14-00688-f004:**
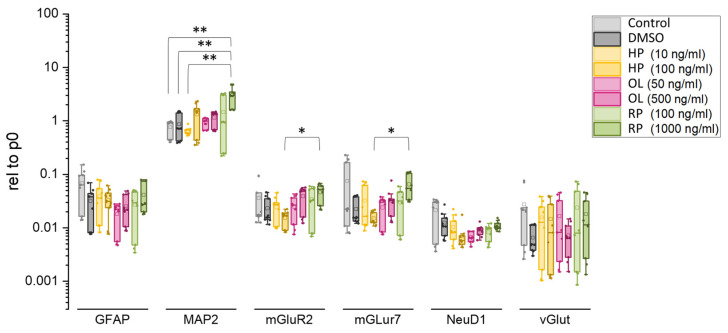
Haloperidol (HP), olanzapine (OL) and risperidone (RP) treatment induced gene expression alterations in the differentiating hippocampal granule cell cultures. * *p* ≤ 0.05 ** *p* ≤ 0.01 Kruskal-Wallis test and post hoc Dunn’s test were done. Data were collected from three biological parallels.

**Figure 5 biomolecules-14-00688-f005:**
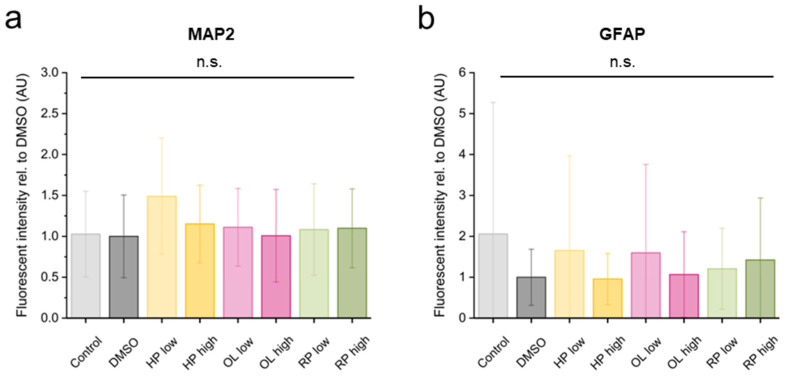
Haloperidol (HP), olanzapine (OL) and risperidone (RP) treatment induced alteration in MAP2 (**a**) and GFAP (**b**) expression in the differentiating hippocampal granule cell cultures. Data were collected from 5 to 10 non-overlapping pictures from each cell culture of 4 independent experiments. For statistical analysis ANOVA was performed with post hoc Tukey test.

**Table 1 biomolecules-14-00688-t001:** The differentiating hippocampal granule cells were treated with haloperidol (HP), olanzapine (OL), and risperidone (RP) at two different concentrations or with dimethyl sulfoxide (DMSO) for 21 days.

Group	Treatment	Concentration	
HP_low_	haloperidol	10 ng/mL	0.003 µM
HP_high_		100 ng/mL	0.03 µM
OL_low_	olanzapine	50 ng/mL	0.16 µM
OL_high_		500 ng/mL	1.6 µM
RP_low_	risperidone	100 ng/mL	0.24 µM
RP_high_		1000 ng/mL	2.4 µM
DMSO	DMSO	0.2 µL/mL	

## Data Availability

The data underlying this article are available on reasonable request.

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
