# Peer review of "Haloperidol, Olanzapine, and Risperidone Induce Morphological Changes in an In Vitro Model of Human Hippocampal Neurogenesis"

_biomolecules, 2024, doi:10.3390/biom14060688_

Round 1

Reviewer 1 Report

Comments and Suggestions for Authors

The paper is suitable for pubblication in the present form

Comments on the Quality of English Language

The paper is suitable for pubblication in the present form

Reviewer 2 Report

Comments and Suggestions for Authors

In the present study, the Authors aimed to investigate how dopamine and serotonin antagonist antipsychotics might affect the proliferation of progenitor cells, neurite development and gene expression of differentiating hippocampal granule cells in vitro.

Overall, I found the present study timely, original, well conducted and scientifically sound. Nevertheless, I have some comments aimed at improving the high quality of the paper, and these are outlined below.

  1. In the introduction, a brief note on the fact that the effects of antipsychotics might be multidimensional, including several targets with different neurobiological underpinnings, should be added with appropriate references (please see and refer to following dois: 10.1080/14740338.2020.1795126 and 10.2174/1570159X22666231128101725).
  2. In the first sentence of the introduction the terms “mental health disorders” are used. Better use “psychiatric disorders” instead.
  3. What were the study limitations and future research directions. Please, add a brief explanation.
  4. Translating into “real world” clinical practice and medicine, what possible clinical shreds of evidence might arise from the present study and what the Researchers do suggest to impact practice in a translational way? Please add a brief résumé paragraph on “implications” in terms of integrative care.

Comments on the Quality of English Language

The English language needs some minor amendments 

Reviewer 3 Report

Comments and Suggestions for Authors

The present manuscript was interesting.

iPSC line from a healthy donor was used in the present study, but how about the iPSC line from psychotic patients?

At the end of the differentiation protocol, approximately 60% of the cells exhibited PROX1 and MAP2 positivity. Did it mean PROX1 and MAP2 double-positive? What was the other 40%? Undifferentiated or glial cells?

Growth curves of NPCs in Figure 2 did not have significant differences among treatments. And, cell counts in Figure 3 were affected. Were the differences between these two data both showing cell numbers based on the condition of measurements, 3 and 10 days culture or used medium?

Reviewer 4 Report

Comments and Suggestions for Authors

The authors have submitted a research article regarding an evaluation by immunocytochemical-based image analysis of a possible morphological changes in an in vitro model of hippocampal dentate gyrus granule cells by application of antipsychotics such as haloperidol, olanzapine, and risperidone. They also examined the expression levels of mRNAs for MAP2 and GFAP as well as their protein levels. The authors demonstrated that a lower concentration of haloperidol significantly induced neuronal differentiation-like phenomenon by evaluating number of processes, outgrowth, arborization, and cell numbers although the expression levels of the essential proteins such as MAP2 and GFAP did not alter, illustrating a hypothesis suggesting that antipsychotics might affect neuronal growth, differentiation, and the death via a possible mechanism of binding by the antipsychotics to multiple cell surface receptors. This issue is of interest, and impact of their results is strong. My overall concern with the article describing the current available data regarding beneficial availability of the evaluation of effects of antipsychotics on neuronal cell growth by using histochemical analysis, offer something substantial that helps advance our understanding of advanced diagnosis and then effective medicinal management available in clinic.

To strengthen authors’ perspectives, the authors are strongly recommended to add a discussion in detail regarding how to evaluate in cases where there is no correlation between the two factors, i.e. image analysis and the protein expression levels. The opposite outcomes, if known, may influence largely the authors’ perspective. Please discuss the effects of a lower but not a higher concentration of haloperidol on the expression levels of the proteins other than MAP2 and GFAP on the basis of the literature.

Round 2

Reviewer 3 Report

Comments and Suggestions for Authors

The present manuscript was well improved and the reviewer understood the authors' responses.

Reviewer 4 Report

Comments and Suggestions for Authors

The authors have done a good job responding to reviewer comments and concerns in their revision. I believe the manuscript is significantly improved as a result. Now I recommend that this revised version of the manuscript can be accepted for publication in Biomolecules.